# Effects of Liquid–Liquid Phase Separation and Relative Humidity on the Heterogeneous OH Oxidation of Inorganic-Organic Aerosols: Insights from Methylglutaric Acid/Ammonium Sulfate Particles

Hoi Ki Lam[1], Rongshuang Xu[1], Jack Choczynski[2], James F. Davies[2], Dongwan Ham[3], Mijung Song[3], Andreas Zuend[4], Wentao Li[5], Ying-Lung Steve Tse[5], Man Nin Chan[1,6]

[1]Earth System Science Programme, Faculty of Science, The Chinese University of Hong Kong, Hong Kong, China

[2]Department of Chemistry, University of California Riverside, Riverside, CA, USA

[3]Department of Earth and Environmental Sciences, Jeonbuk National University, Jeollabuk-do, Republic of Korea

[4]Department of Atmospheric and Oceanic Sciences, McGill University, Montreal, Québec, Canada

[5]Departemnt of Chemistry, The Chinese University of Hong Kong, Hong Kong, China

[6]The Institute of Environment, Energy and Sustainability, The Chinese University of Hong Kong, Hong Kong, China

Corresponding author: mnchan@cuhk.edu.hk

## Abstract

Organic compounds residing near the surface of atmospheric aerosol particles are exposed to chemical reactions initiated by gas-phase oxidants, such as hydroxyl (OH) radicals. Aqueous droplets composed of inorganic salts and organic compounds can undergo phase separation into two liquid phases, depending on aerosol composition and relative humidity (RH). Such phase behavior can govern the surface characteristics and morphology of the aerosols, which in turn affect the heterogeneous reactivity of organic compounds toward gas-phase oxidants. In this work, we used an aerosol flow tube reactor coupled with an atmospheric pressure ionization source (Direct Analysis in Real Time) and a high-resolution mass spectrometer to investigate how phase separation in model aqueous droplets containing an inorganic salt (ammonium sulfate, AS) and an organic acid (3-methyglutaric acid, 3-MGA) with an organic-to-inorganic dry mass ratio (OIR) of 1 alters the heterogeneous OH reactivity. At high RH, 3-MGA/AS aerosols were aqueous droplets with a single liquid phase. When the RH decreased, aqueous 3-MGA/AS droplets underwent phase separation at ~75 % RH. Once the droplets were

phase-separated, they exhibited either a core–shell, partially engulfed, or a transition from core–shell to partially engulfed structure, with an organic-rich outer phase and an inorganic-rich inner phase. The kinetics, quantified by an effective heterogenous OH rate constant, was found to increase gradually from $1.01 \pm 0.02 \times 10^{-12}$ to $1.73 \pm 0.02 \times 10^{-12}$ cm$^3$ molecule$^{-1}$ s$^{-1}$ when the RH decreased from 88 % to 55 %. The heterogeneous reactivity of phase-separated droplets is slightly higher than that of aqueous droplets with a single liquid phase. This could be explained by the finding that when the RH decreases, higher concentrations of organic molecules (i.e. 3-MGA) are present at or near the droplet surface, which are more readily exposed to OH oxidation, as demonstrated by phase separation measurements and model simulations. This could increase the reactive collision probability between 3-MGA molecules and OH radicals dissolved near the droplet surface and secondary chain reactions. Even for phase-separated droplets with a fully established core–shell structure, the diffusion rate of organic molecules across the organic-rich outer shell is predicted to be fast in this system. Thus, the overall rate of reactions is likely governed by the surface concentration of 3-MGA rather than a diffusion limitation. Overall, understanding the aerosol phase state (single liquid phase versus two separate liquid phases) is essential to better probe the heterogenous reactivity under different aerosol chemical composition and environmental conditions (e.g. RH).

## 1. Introduction

Organic compounds present at or near the surface of atmospheric aerosols can be efficiently reacted with gas-phase oxidants, such as OH, ozone ($O_3$), and nitrate radicals (George and Abbatt, 2010; Kroll et al., 2015; Chapleski et al., 2016), continuously changing aerosol properties (e.g. water uptake and cloud formation potential) through the alteration of surface and bulk composition (Lambe et al., 2007; Cappa et al., 2011; Harmon et al., 2013; Slade et al., 2015, 2017). The physical state of aerosol particles (and their separate phases, when present) is known to play a role in the heterogeneous reactivity of organic aerosols (Ruehl et al., 2013; Chan et al., 2014; Slade and Knopf, 2014; Li et al., 2020). However, the heterogeneous reactivity of aqueous droplets containing organic compounds and inorganic salts remains unclear. Additional uncertainties in heterogeneous reactivity of organic compounds arise when these droplets undergo phase separation. Recent studies suggest that these organic–inorganic droplets can undergo liquid–liquid phase separation (LLPS) depending on environmental conditions (e.g. RH and temperature), and aerosol composition (including different types of inorganic salts, the average oxygen-to-carbon (O:C) elemental ratio of organic compounds, and OIR) (Bertram et al., 2011; Zuend and Seinfeld, 2013; You et al., 2014; Qiu and Molinero,

2015; Charnawskas et al., 2017; Losey et al., 2018; Freedman, 2020). These phase-separated droplets exhibit two distinct liquid phases: typically, an inorganic-rich inner phase and an organic-rich outer phase as depicted in **Figure 1**. Different morphologies have been observed, for instance, core–shell morphology, partially engulfed morphology and transitions between different types of morphology. Such complex phase behavior has been found to influence the distribution of black carbon in the aerosols (Brunamonti et al., 2015), water uptake of the aerosols (Chan and Chan, 2007; Hodas et al., 2015), reactive uptake of gas-phase species (e.g. dinitrogen pentoxide ($N_2O_5$) and dimethylamine (($CH_3)_2NH$)) (Gaston et al., 2014; DeRieux et al., 2019), and the gas–particle partitioning of internally mixed organic–inorganic aerosols (Zuend and Seinfeld, 2012; Shiraiwa et al., 2013). To date, it remains unclear to what extent LLPS will alter the heterogeneous oxidation of aqueous organic–inorganic droplets under different composition and relative humidity conditions.

The phase separation behavior could vary with the hydrophilicity of atmospheric organic compounds (or hydrophobicity). While the state of two separate liquid phases is the dominant aerosol phase state for hydrophobic organic compounds, a single liquid phase and two separate liquid phases are both important aerosol phase states for atmospheric aerosols with oxygenated organic compounds. Studies have demonstrated that aqueous organic–inorganic droplets containing a single or a mixture of oxygenated organic compounds (e.g. polyols and carboxylic acids) may undergo phase separation to form an organic-rich outer phase; typically, this transition occurs at a RH below 90 % because of the moderate solubility of organics in aqueous ionic solutions. Previous work has explored the heterogeneous reactivity of aqueous organic–inorganic droplets with hydrophobic organic shells (or coatings) (e.g. McNeill et al., 2007; Dennis-Smither et al., 2012). However, the effects of phase separation on reactivity in aqueous organic–inorganic droplets containing moderately oxygenated organic compounds remain largely unexplored.

In a previous study, we have investigated the heterogeneous OH oxidation of aqueous droplets containing 3-methylglutaric acid (3-MGA) and AS with an OIR of 2 at a relatively high RH of 85 % (Lam et al., 2019). AS is a typical inorganic salt in atmospheric aerosols and was chosen to analyze the salting-in/out effect on the heterogeneous reactivity. 3-MGA was chosen as a model compound for small branched carboxylic acids, because 3-MGA and its structural isomer (2-methylglutaric acid, 2-MGA) are two of the most abundant methyl-substituted dicarboxylic acids that have been detected in tropospheric aerosols (Kawamura and Kalpan,

1987; Li et al., 2015; Kundu et al., 2016). Furthermore, these small branched dicarboxylic acids can induce liquid–liquid phase separation in aqueous inorganic-organic droplets when they are mixed with inorganic salts (Losey et al., 2016). These studies suggest that small, branched diacids could play a role in determining both the occurrence and compositional extent of phase separation as well as the heterogeneous reactivity of aqueous organic–inorganic droplets. We found that aqueous 3-MGA/AS droplets with OIR of 2 become initially phase separated at an RH of ~ 73 %. During oxidation, aqueous 3-MGA/AS droplets were expected to remain in a single liquid phase, since the RH inside the reactor was above the separation RH (SRH). The presence of dissolved ions (i.e. $NH_4^+$ and $SO_4^{2-}$ from AS and some amount of bisulfate ($HSO_4^-$)) does not substantially alter the reaction mechanisms, but slows down the overall rate of reactions with OH as compared to that of pure 3-MGA droplets. However, at lower RH, droplets with two separate liquid phases might have an OH reactivity different from those with a single liquid phase at higher RH, since the surface composition of the droplet changes.

To better understand the role of aerosol phase state (single liquid phase versus two separate liquid phases) on the heterogeneous reactivity, we investigated how the phase separation in aqueous droplets containing an inorganic salt (AS) and an organic compound (3-MGA) with an OIR of 1 alters the kinetics and products upon heterogeneous OH oxidation (**Table 1**). First, the separation RH (SRH; the LLPS onset RH during dehumidification) and morphologies of phase-separated 3-MGA/AS droplets were determined using two distinct techniques: an optical microscopy setup and a linear-quadrupole electrodynamic balance. Second, the molecular composition of 3-MGA/AS droplets before and after OH oxidation was investigated using an aerosol flow tube reactor coupled with a Direct Analysis in Real Time (DART) ionization source and a high-resolution mass spectrometer at different RH levels. Based on these results, we attempt to explore and analyze quantitatively how phase separation in aqueous organic–inorganic droplets may determine the heterogeneous OH reactivity of methyl-substituted dicarboxylic acids.

## 2. Experiment

### 2.1 Phase separation measurements

#### 2.1.1 Phase separation measurement of aqueous 3-MGA/AS droplets using optical microscopy

Aqueous solutions containing 3-MGA and AS with OIR = 1 were first nebulized onto Teflon substrates (Alphaflon, Korea). The Teflon slides containing 3-MGA and AS were then placed into a flow-cell coupled to an optical microscope (Olympus BX43, 40X objective). The

detailed experimental methods and procedures are described in Song et al. (2017, 2018). In brief, the RH inside the flow-cell was adjusted by varying the mixing ratios of $N_2$ and water vapor. The RH was measured by a combined humidity and temperature sensor (Sensirion, Switzerland), which has been calibrated by measuring the known deliquescence relative humidities of aqueous inorganic droplets covering a RH range from ~44 % to 94 % (e.g. AS, sodium chloride, potassium carbonate and potassium nitrate). The accuracy of the sensor was ± 1.5 % RH with the calibration. For the measurements, micrometer-sized droplets were first equilibrated at a RH close to 100 %. The RH was then decreased continuously by 0.2 − 0.3 % RH/min. The measurements were stopped when (partial) efflorescence of the droplets was observed. Optical images of the aqueous droplets were recorded every 5 seconds using a metal–oxide–semiconductor sensor (DigiRetina 16, Tucsen, China). The flow cell was kept at a controlled temperature of 290 ± 1 K.

### 2.1.2 Phase separation measurement of aqueous 3-MGA/AS droplets using a linear-quadrupole electrodynamic balance

A linear-quadrupole electrodynamic balance (LQ-EDB) was used to levitate micron-sized droplets containing 3-MGA and AS (OIR = 1) under controlled RH conditions. The experimental platform has been introduced elsewhere (Davies et al., 2019), and a brief description is given in the following. A droplet was prepared from a dilute solution using a microdroplet dispenser, given a charge (on the order of 50 fC) and was introduced into the LQ-EDB where it became confined by the electric field and suspended indefinitely. The droplet was illuminated with a 532 nm laser and a broadband LED (660 nm with 20 nm line-width). The elastic light scattering pattern was imaged with a camera and the Mie resonance spectrum was measured using an Ocean Optics HR4000+ spectrometer (Price et al., 2020). The size and refractive index of the droplet were determined from the peak positions in the resonance spectrum analyzed using the algorithms of Preston et al. (Preston and Reid, 2015). The RH was controlled through introduction of a mixture of dry and humidified nitrogen and measured using a calibrated capacitance probe (Honeywell HIH-4602-L). As the RH was lowered, changes in the size and RI were measured. Phase separation was identified by a discontinuity in the size or RI, or a loss of clear resonance peaks, associated with a loss of spherical symmetry (Stewart et al., 2015; Gorkowski et al., 2016).

### 2.2 Heterogeneous OH oxidation of 3-MGA/AS droplets over a range of RH

An atmospheric pressure aerosol flow tube reactor was used to investigate the heterogeneous

OH oxidation of 3-MGA/AS droplets (OIR = 1) at different RH (55 to 88 % RH) and 20 °C. The details of the experiments can be found in previous work (Lam et al., 2019) and is briefly described below. First, aqueous 3-MGA/AS droplets with an OIR of 1 were generated by atomization. The droplet stream (without passing through a diffusion dryer) was then introduced into the reactor together with $O_3$, oxygen ($O_2$) and a tracer gas (hexane). The RH was changed by varying mixing ratios of dry $N_2$ and a $N_2/H_2O$ gas flow.

Inside the reactor, the formation of OH radicals follow the photolysis of $O_3$ by UV light at 254 nm. This forms excited $O(^1D)$ atoms, some of which react with water molecules ($H_2O$) to generate OH radicals. The concentration of OH radicals was measured *in-situ* from the decay of hexane, which was traced by a gas chromatograph and was used to compute the OH exposure defined as the product of OH concentration and residence time (1.3 min.) using a relative rate method (Smith et al., 2009). The maximum OH exposure was about $8.4$–$8.9 \times 10^{11}$ molecules $cm^{-3}$ s. After removing the gas-phase species and $O_3$ remaining in the stream leaving the reactor using a Carulite denuder and a charcoal denuder, the aerosol size distribution was measured by a scanning mobility particle sizer (SMPS, TSI). For *in situ* aerosol chemical characterization, the aerosols were fully vaporized inside an aerosol heater. The gas-phase species were then introduced into an ionization region, which is a small open space between the DART ionization source (DART SVP, IonSense Inc.) and an atmospheric inlet of a high-resolution mass spectrometer (Q Exactive Orbitrap Mass Spectrometer, ThermoFisher) (Nah et al., 2013; Chan et al., 2013). In the ionization region, anionic oxygen ions ($O_2^-$) can be produced via reactions between atmospheric $O_2$ molecules and electrons generated by the Penning ionization of metastable helium (He) atoms in the ionization source (Cody et al., 2005). $O_2^-$ can then react with gas-phase species (M) by proton abstraction to produce deprotonated molecular ions ([M–H]$^-$), sampled by the mass spectrometer. Mass spectra were collected at a resolution of about 140,000 and analyzed using Xcalibur software.

## 3. Results and Discussion

### 3.1 Phase separation and morphology measurements

*Optical microscopy:* Upon dehumidification, all droplets containing 3-MGA/AS underwent LLPS on a hydrophobic substrate at $75.2\pm0.4$ % RH showing configurations characteristic of either core–shell, partially engulfed, or a change of morphology from core–shell to partially engulfed. Out of 25 different 3-MGA/AS droplets studied, 16 droplets showed a change from

core–shell to partially engulfed morphologies, 6 droplets showed partially engulfed, and 3 droplets showed core–shell morphologies over the observed RH range from the onset of phase separation to the efflorescence of AS within the droplets. As shown in **Figure 2**, at high RH, aqueous droplets exhibited a single liquid phase. As the RH decreased, the formation of many small inclusions, which likely contain dissolved AS and water, was observed at ~75 % RH, suggesting the onset of phase separation. The phase separation mechanism involved is very likely spinodal decomposition (Papon et al., 2006; Song et al., 2012ab). After onset of LLPS, the droplets displayed different morphologies, some showing a shift of morphology from core–shell to partially engulfed configurations (**Figure 2a**), partially engulfed morphologies (**Figure 2b**), and core–shell morphologies (**Figure 2c**). Interestingly, for most of the 3-MGA/AS droplets, a morphological transition from core–shell to partially engulfed was observed, with the inner phase moving toward the edge of the droplet at RH ranges from ~73% to ~67% during dehumidification (**Figure 2a**). The formation of different morphologies during phase separation has been reported in the literature. For instance, both core–shell, partially engulfed morphologies, and even a shift of morphology from core–shell to partially engulfed morphologies have been observed for $C_6$-diacids/AS droplets (Song et al., 2012ab; 2013). These observations could be explained by a relatively high sensitivity of the morphologies of phase-separated droplets deposited on the hydrophobic substrate with regard to small variations in actual phase volume ratios, aerosol compositions, surface tension, and interfacial tension with the substrate and among liquid phases, resulting in different spreading coefficients (Kwamena et al., 2010; Reid et al., 2011; Song et al., 2013; Stewart et al., 2015)

*LQ-EDB:* Phase separation of aqueous 3-MGA/AS droplet was also investigated by levitating a single droplet with diameter of ~4 μm in a LQ-EDB. **Figure 3a** shows the RH dependence of radius and refractive index (RI) of the droplet as the RH was decreased stepwise in 0.5 % increments from 83 % down to < 70 % RH. At around 73 % RH, there is a discontinuity in the measured data and a subsequent change in the slope of RI vs. RH. This discontinuity is more clearly identified in **Figure 3b**, showing the RI as a function of droplet radius. At around 3850 nm, the RI increases much more sharply as the droplet size decreases. This change in behavior is likely attributed to LLPS to form a core–shell morphology. The resonance spectrum appears virtually unchanged due to the spherical symmetry of the LLPS state (**Figure 3c**). Measurements on larger droplets around this RH also show signatures of a core–shell morphology. In **Figure S1**, the RI established using $3^{rd}$ and $4^{th}$ order modes are compared to

the results when using 3$^{rd}$ order modes only. A large difference in the RI is observed, indicating that when 4$^{th}$ order modes are present, which penetrate deeper into the droplet, the liquid–liquid phase boundary is crossed resulting in erroneous analysis using a method that assumes the RI to be homogeneous throughout the sample. The core–shell morphology is lost at lower RH and the size and RI cannot be retrieved below 68 % RH. The spectra recorded after this point are shown in **Figure 3c**, with a clear loss of sharp resonant peaks. This could be explained by the loss of spherical symmetry, indicating the formation of a partially engulfed morphology.

Generally, the phase separation behaviors of different aqueous 3-MGA/AS droplets determined by the two methods agree well. We acknowledge that the phase separation and droplet morphology characteristics studied here using the optical microscopy setup and the single particle levitation method were both conducted on super-micron-sized droplets; the phase separation of submicron-sized 3-MGA/AS droplets has not been measured directly. O'Brien et al. (2015) reported that the phase separation behavior (i.e. SRH) of droplets with diameter of ~1 μm are very similar to that with diameter larger 20 μm. Krieger et al. (2012) suggest that aerosol size could potentially play a role in the phase separation behavior when the aerosol diameter is smaller than ~ 100 nm and the Kelvin effect becomes more significant. Moreover, recent modeling work and cryogenic-transmission electron microscopy studies show that phase separation can also be observed down to the particle sizes with diameters of a few tens of nm to about a hundred nm, depending on experimental protocol (e.g. drying rate) (Veghte et al., 2013, 2014; Altaf et al., 2016; Altaf and Freedman 2017). These results support that the phase separation behavior reported for micron-sized droplets provides a good first approximation for aerosol size range (~200 nm) used for heterogeneous oxidation in this work.

*3.2 Thermodynamic phase equilibrium calculations*

Thermodynamic phase equilibrium calculations were also performed using the Aerosol Inorganic–Organic Mixtures Functional groups Activity Coefficients (AIOMFAC) liquid–liquid equilibrium (LLE) model, hereafter referred to as AIOMFAC-LLE model, to compare the results of the experimentally observed LLPS range and onset mechanism (Zuend et al., 2008, 2010, 2011; Zuend and Seinfeld, 2013). As shown in **Figure 4**, 3-MGA/AS droplets with OIR = 1 correspond to a dry mixture mass fraction of AS of $w_d$(salt) = 0.5 in the phase diagram. Lowering of RH (or water activity, $a_w$) at $w_d$(salt) = 0.5 will lead to a composition trajectory entering the phase-separated composition space by almost immediately crossing the spinodal (dotted) curve, thereby entering the intrinsically unstable area of the composition space (blue

region). These model predictions are in strong support of the observation of spinodal decomposition as the likely mode of LLPS for this OIR during dehumidification experiments. Although the onset RH of LLPS predicted by the model occurs at an RH of ~83 %, about 8 % higher in RH than in the microscopy experiments. This is not uncommon for a predictive model like AIOMFAC-LLE, since this group-contribution model was not specifically tuned to perform accurately for the organic–inorganic system studied here. Based on this comparison and other related comparisons of AIOMFAC-LLE predictions with measurements (Song et al., 2012a), we estimate that the onset of LLPS is predicted within about 10 % uncertainty in RH.

**Figure S2** shows the simulated equilibrium aerosol composition for 3-MGA/AS droplets (OIR = 1) at 293 K with RH ranging from 30 % to 99 %, as a first approximation. After the aqueous droplet underwent phase separation, it is predicted to consist of two separate liquid phases: a salt-rich inner phase (phase α) and an organic-rich outer phase (phase β) (**Table S1**). When phase separation occurs at ~ 83 %, a non-negligible amount of 3-MGA is predicted to be present in the salt-rich phase for RH > 74 %. At lower RH (RH < 74 %), 3-MGA remaining in the salt-rich phase partitions to the organic-rich phase, resulting in a negligible amount of 3-MGA in the salt-rich phase. As the water content in the aerosol has an inverse relationship with RH for both phases, the mass fractions of both 3-MGA and AS increase as RH decreases (prior to any crystallization). Assuming a core–shell structure, the model can predict the equilibrium radial thicknesses of both phases of phase-separated droplets with radius ranging from 85.5 to 91.5 nm at different RH. The radial thickness of the organic-rich outer phase was predicted to increase slightly from 23.6 nm to 24.9 nm as the RH decreases from 70 % to 55 % (**Table S1**).

The viscosity of 3-MGA/AS droplet with a single liquid phase (RH > SRH) and the organic-rich phase of phase-separated 3-MGA (RH < SRH) were also estimated using the AIOMFAC-VISC model (Gervasi et al. 2020). As shown in **Figure S3**, when the RH > SRH, the viscosity of an aqueous 3-MGA/AS droplet with a single liquid phase ranges from $\sim 10^{-3} - 10^{-2}$ Pa s and increases with decreasing RH. This is mainly attributed to the ability of water to act as a plasticizer, i.e. a component that reduces the viscosity of a solution (Koop et al., 2011). On the other hand, when the RH < SRH, for the phase-separated droplet, a more drastic change in the viscosity of the organic-rich phase is observed when the RH varies. The viscosity of the organic-rich phase is predicted to increase from $2.8 \times 10^{-2}$ to $9.6 \times 10^{-2}$ Pa s when the RH decreases from 70 % to 55 %. As a first approximation, these aerosol composition and viscosity data predicted by the two models will be used in the following sections to examine how the

phase separation and RH would alter the heterogeneous reactivity.

### 3.3 Aerosol-DART mass spectra of 3-MGA/AS aerosols

**Figure S4** shows the aerosol-DART mass spectra over a range of RH. Very similar mass spectra were observed at different RH. Before oxidation, two major peaks at $m/z$ = 97 (bisulfate ion,
$HSO_4^-$) and $m/z$ = 145 (deprotonated molecular ion of 3-MGA, $C_6H_9O_4^-$) are observed. $HSO_4^-$ likely originates from the AS (Lam et al., 2019). Peak at $m/z$ = 143 has a chemical formula of $C_6H_7O_4^-$ ion and is likely a background ion or impurity. After oxidation, the formation of $C_6$ ketone and $C_6$ hydroxyl functionalization products ($C_6H_7O_5^-$, $m/z$ = 159 and $C_6H_9O_5^-$, $m/z$ = 161) were observed. Some small product peaks with less than 3% of the total ion signal, such
as $C_4H_5O_3^-$, $C_5H_5O_3^-$, $C_5H_7O_3^-$, $C_5H_7O_4^-$, $C_6H_5O_5^-$ and $C_6H_7O_6^-$, are also detected. These results are consistent with the peaks observed in our previous work (Lam et al., 2019).

The heterogeneous OH oxidation of aqueous 3-MGA/AS aerosols, consisting of either a single liquid phase (RH > SRH) or two separate liquid phases (RH < SRH), was investigated by
varying the RH. Since the aerosol-DART mass spectra measured at different RH are about the same, they suggest that changes in aerosol phase (i.e. single liquid phase or two separate liquid phases) and composition (e.g. concentration of 3-MGA, dissolved ions and water) in response to the change in RH do not alter the reaction mechanisms. While the reaction pathways have been discussed previously (Lam et al., 2019), we here primarily focus on the effects of phase
separation on heterogeneous oxidation kinetics.

### 3.4 Oxidation kinetics of 3-MGA/AS aerosols

To quantify the kinetics, the effective heterogeneous OH rate constant ($k$) can be obtained at different RH through fitting the normalized decay of 3-MGA (($I/I_0$)) (**Figure 5**) at different
OH exposure:

$$\ln\frac{I}{I_0} = -k[\text{OH}]t \tag{1}$$

where $I_0$ and $I$ are the ion signals of 3-MGA before oxidation and after oxidation, respectively, [OH] $t$ represents the OH exposure. Based on the kinetic data measured at different RH (**Figure 5**), we will first discuss the effects of phase separation on the heterogeneous OH reactivity of
3-MGA (Section 3.4.1). We will then further discuss the RH effects on the heterogeneous OH reactivity of aqueous droplets with a single liquid phase (RH > SRH) (Section 3.4.2) and two separated liquid phases (RH < SRH) (Section 3.4.3).

*3.4.1 Effect of phase separation on the heterogeneous reactivity: Single liquid phase vs. two*

*separate liquid phases*

To investigate how the phase separation affects the kinetics, we attempt to compare the heterogenous reactivity of aqueous droplets with a single liquid phase (RH > SRH) to that of phase-separated droplets (RH < SRH). The reaction rate ($k$) values are used instead of the effective OH uptake coefficient, $\gamma_{eff}$, since different morphologies that can be adopted by phase-separated droplets (either a core–shell or partially engulfed structure with multiple inclusions as shown in **Figure 2 and Figure 3**) would introduce uncertainties in the estimation of $\gamma_{eff}$, which would require prior knowledge of a well-defined aerosol morphology (Davies and Wilson, 2015). We also acknowledge that since polydisperse aerosols (narrow size range) were investigated in our study, the spread of aerosol size could affect the determination of $k$. However, as shown in **Figure S5**, before oxidation, the mean surface-weighted aerosol diameters at different RH are about the same, ranging from about 196 to 199 nm. The change in aerosol size upon oxidation was not very significant. Hence, the effect of aerosol size on the kinetics is expected to be insignificant (particularly within the rather small diameter range investigated).

As shown in **Figure 5**, when the RH decreases from 88 % to 55 %, the $k$ value increases from $1.01 \pm 0.02 \times 10^{-12}$ to $1.73 \pm 0.02 \times 10^{-12}$ cm$^3$ molecule$^{-1}$ s$^{-1}$. This suggests that the heterogeneous OH reactivity of 3-MGA in phase-separated droplets (RH < SRH) is slightly higher than that of aqueous droplets with a single liquid phase (RH > SRH). For instance, the rate constants of phase-separated droplets range from $1.34 \pm 0.03 \times 10^{-12}$ to $1.73 \pm 0.02 \times 10^{-12}$ cm$^3$ molecule$^{-1}$ s$^{-1}$ while those of aqueous droplets with a single liquid phase range from $1.01 \pm 0.02 \times 10^{-12}$ to $1.28 \pm 0.02 \times 10^{-12}$ cm$^3$ molecule$^{-1}$ s$^{-1}$. This could be explained by the inhomogeneous distribution of 3-MGA within the droplets created by LLPS, with higher concentrations present at or near the droplet surface, as demonstrated by phase separation measurements and model simulations. This could increase the probability that 3-MGA molecules collide with OH radicals at or near the aerosol surface, leading to a higher overall rate of reactions. For instance, Xu et al. (2020) have performed molecular dynamics (MD) simulations to demonstrate that upon heterogeneous reaction, a gas-phase OH radical has to be first absorbed by the aerosol, and an absorbed OH radical would then require to collide with an organic molecule many times by diffusion before the reaction occurs at the aerosol surface. A higher concentration of organic molecules could reduce the distance between the OH radical

and its neighboring organic molecules. This would increase the heterogeneous reactivity, which is consistent with the enhanced reaction rates observed for phase separated droplets relative to aqueous droplets with a single liquid phase.

Previous studies have revealed that the alkoxy radical are likely generated from the hydrogen abstraction at the tertiary carbon upon the heterogeneous OH oxidation of methyl-substituted dicarboxylic acids (e.g., 2-MGA, 3-MGA and 2,3-dimethylsuccinic acid) and could play a significant role in governing the heterogeneous kinetics and chemistry (Cheng et al., 2015;

Chim et al., 2017; Lam et al., 2019). Upon oxidation, 3-MGA molecules ($C_6H_{10}O_4$) can undergo hydrogen abstraction by OH radicals, producing alkyl radicals ($C_6H_9O_4\bullet$) which quickly react with $O_2$ to yield peroxy radicals ($C_6H_9O_6\bullet$) (**R1** and **R2**). Peroxy radicals are more likely to produce alkoxy radicals ($C_6H_9O_5\bullet$) through self-reactions (**R3**), which can abstract hydrogen atoms from neighboring organic molecules (e.g. unreacted 3-MGA,

$C_6H_{10}O_4$) (**R4**). This allows the formation of stable alcohol products ($C_6H_{10}O_5$) and alkyl radicals ($C_6H_9O_4\bullet$) which can produce peroxy radicals ($C_6H_9O_6\bullet$) eventually. As a result, alkoxy radicals can once again be produced via self-reactions of peroxy radicals and help propagate the chain reactions. In this work, it is likely that when a phase separation occurs, a higher concentration of 3-MGA in the organic-rich outer phase of phase-separated droplets

could favor these secondary reactions, increasing the reaction rate.

$$C_6H_{10}O_4 + OH\bullet \rightarrow C_6H_9O_4\bullet + H_2O \tag{R1}$$
$$C_6H_9O_4\bullet + O_2 \rightarrow C_6H_9O_6\bullet \tag{R2}$$
$$C_6H_9O_6\bullet + C_6H_9O_6\bullet \rightarrow 2\ C_6H_9O_5\bullet + O_2 \tag{R3}$$
$$C_6H_9O_5\bullet + C_6H_{10}O_4 \rightarrow C_6H_9O_4\bullet + C_6H_{10}O_5 \tag{R4}$$

*3.4.2 The effect of RH on the heterogeneous reactivity of aqueous 3-MGA/AS droplets with a single liquid phase (RH > SRH)*

When the RH is above the determined SRH, 3-MGA/AS aerosols are likely aqueous droplets

with a single liquid phase (**Figure 1**). As shown in **Figure 4**, when the RH decreases from 88 % to 75 %, the determined $k$ increases slightly from $1.01 \pm 0.02 \times 10^{-12}$ to $1.28 \pm 0.01 \times 10^{-12}$ $cm^3$ $molecule^{-1}$ $s^{-1}$. With prior knowledge of aerosol morphology, to better quantify the effect of RH on the oxidation kinetics of aqueous droplets with a single liquid phase, the $k$ value can be used to compute the effective OH uptake coefficient, $\gamma_{eff}$ (Davies and Wilson, 2015):

$$\gamma_{eff} = \frac{2\rho_0 D_0 m_f\ N_A k}{3 M_w \overline{c_{OH}}}. \tag{2}$$

Here, $\rho_0$ is the aerosol density, $D_0$ is the mean surface-weighted aerosol diameter, $m_f$ is the mass fraction of 3-MGA in the aerosol, $N_A$ is Avogadro's number, $M_w$ is the molar mass of 3-MGA and $\overline{c_{OH}}$ is the thermal speed of OH radicals. The aerosol diameter before OH oxidation ranges from 196 nm to 199 nm for 75–88 % RH (**Figure S5**). The approximate mass fraction of 3-MGA ($m_f$) is obtained from equilibrium composition calculations performed by the AIOMFAC-LLE model (**Table S1**). The density of the droplets was computed based on the aerosol composition simulated using the volume additivity rule with pure liquid-state densities of the species. Using **Eqn. 2**, the $\gamma_{eff}$ at 75 %, 80 %, 85 % and 88 % RH are calculated to be $0.40 \pm 0.02$, $0.33 \pm 0.02$, $0.25 \pm 0.01$ and $0.20 \pm 0.01$, respectively (**Table 1**). The reactivity is found to decrease as the RH increases. This could be attributed to the dilution effect (Davies and Wilson, 2015). For instance, as shown in **Table S1**, the mass fraction of 3-MGA is shown to decrease from 0.292 to 0.197 when the RH increases from 75 % to 88 % RH. As a result of lower concentration of 3-MGA near the aerosol surface, 3-MGA molecules will be less likely to collide with OH radicals impinging the aerosol surface and the secondary reactions are expected to be lowered, as discussed in section 3.4.1

We would like to note that at 85 % RH, $\gamma_{eff}$ for pure 3-MGA ($\gamma_{eff} = 2.41 \pm 0.13$) and aqueous 3-MGA/AS droplets with OIR of 2 ($\gamma_{eff} = 0.99 \pm 0.05$) measured in our previous work (Lam et al., 2019), are higher than our results for the OIR = 1 case determined here ($\gamma_{eff} = 0.25 \pm 0.01$). $\gamma_{eff}$ greater than 1 reported in the literature suggests that secondary chain reactions are likely to occur upon oxidation. This may be attributed to the importance of the alkoxy radical chemistry. At a given RH (i.e. 85 % RH), when the amount of inorganic salts (e.g. AS) dissolved in the droplets increases (i.e. decreasing OIR), the concentration of organic compounds decreases. This could hinder the secondary chain reactions initialized by alkoxy radicals, slowing down the overall reaction rate.

*3.4.3 The effect of RH on the heterogeneous reactivity of aqueous droplets with two separate liquid phases (RH < SRH)*

For phase-separated 3-MGA/AS droplets, the *k* value has been found to increase slightly from $1.34 \pm 0.03 \times 10^{-12}$ to $1.73 \pm 0.02 \times 10^{-12}$ cm$^3$ molecule$^{-1}$ s$^{-1}$ when the RH decreases from 70 % to 55 % (**Table 1**). These values suggest that higher reactivities are observed at lower RH. For phase-separated droplets, the partitioning between an inorganic-rich inner phase and an organic-rich outer phase associated with RH would be expected to show a higher 3-MGA

concentration in the organic-rich outer phase at lower RH, especially in the case of a core–shell morphology (**Table S1**). Similar to aqueous 3-MGA/AS droplets with a single liquid phase at high RH (RH > SRH), the overall rate of OH reaction with 3-MGA could be enhanced by 1) the reactive collision frequency between 3-MGA molecules and OH radicals near the aerosol surface and 2) the secondary chain reactions initialized by alkoxy radicals, which likely become more favorable at higher solute concentrations within the outer organic shell at lower RH.

For phase-separated droplets, the diffusivity of organic molecules across the organic-rich outer phase has been suggested to influence the heterogeneous kinetics (Zhou et al., 2019). Additionally, when the organic-rich phase becomes more viscous at lower RH, the oxidation can be limited by the slow diffusion (Davies and Wilson, 2015). We carried out a simple analysis to estimate the time scale for diffusive mixing of 3-MGA within the organic-rich phase to investigate whether the molecular diffusion of organic molecules (i.e. 3-MGA) across the organic-rich phase (outer shell) controls the heterogeneous reactivity of phase separated droplets with a fully developed core–shell structure. The diffusion coefficient ($D$) of the organic molecules can be estimated from the AIOMFAC-VISC model prediction of viscosity (**Table S1**) using the Stokes–Einstein relation (Koop et al, 2011; Gržinić et al., 2015; Steimer et al., 2015):

$$D = \frac{k_B T}{6\pi\eta r} \tag{3}$$

where $k_B$ is the Boltzmann constant, $T$ is the temperature, $\eta$ is the dynamic viscosity and $r$ is the radius of a 3-MGA molecule, which is estimated using the molecular weight and density of 3-MGA (Lam et al., 2019). The diffusion coefficient of the organic-rich phase of phase-separated droplets is estimated to range from $6.2 \times 10^{-12}$ to $2.1 \times 10^{-11}$ m$^2$ s$^{-1}$ for the RH range of 55 % to 70 %. The time scale for diffusive mixing ($\tau$) can then be estimated from the diffusion coefficient, $D$ as (Gržinić et al., 2015; Steimer et al., 2015):

$$\tau = \frac{R^2}{D\pi^2} \tag{4}$$

Here R is the characteristic mixing length. Using the estimates of radial shell thickness (**Table S1**), the diffusive mixing time scale within the organic-rich outer shell is estimated to increase from 2.6 μs to 10.2 μs when the RH decreases from 70 % to 55 %. The time between two successive collision events between gas-phase OH radical and the aerosol surface, $\tau_{coll}$, is estimated to be 4.8 μs at the highest OH radical concentration investigated in the work (i.e. the maximum OH exposure) using **Eqn. 5** (Chim et al., 2017; 2018):

$$\tau_{coll} \cong \frac{4}{[OH]\,\overline{c_{OH}}\,A} \qquad (5)$$

Here *A* is the surface area of the droplets. Under our experimental conditions, although these diffusive mixing time scales ($\tau$) are not always smaller than the time scale for the collision between gas-phase OH radicals and the aerosol surface ($\tau_{coll}$), the fast diffusion likely allows for sufficient mixing within the time scale of reaction events throughout the RH range investigated. It might be reasonable to assume that the oxidation is not likely to be controlled by the diffusion of organic molecules across an organic-rich outer shell of phase-separated 3-MGA/AS droplets. It also notes that ambient gas-phase OH radical concentration is lower than that used in this study. This suggests that the species would have more time to diffuse to the aerosol surface for oxidation. Hence, the overall rate of the oxidation will be less likely limited by the diffusion at lower gas-phase OH radical concentrations in the atmosphere. We acknowledge that multiple inclusions were observed in phase-separated 3-MGA/AS droplets during the phase separation measurement with optical microscopy on micron-sized droplets (**Figure 1**). The effects of these inclusions on the diffusivity of 3-MGA in the organic-rich phase and from droplet interior to the surface are unknown at present. For instance, the AS inclusions in contact with the droplet/air interface of partially engulfed 3-MGA may possibly affect the reactivity of 3-MGA by changing the surface coverage and concentration of 3-MGA exposed to the gas phase. Furthermore, the presence of inclusions may affect the diffusion path of organic molecules to the surface (where oxidation predominantly occurs). Hence, future investigations in quantifying the effects of different morphologies on molecular orientation and heterogeneous reactivity may be warranted since the morphology of phase-separated aerosols can be sensitive to the aerosol composition.

*3.4.4 A molecular dynamics perspective on 3-MGA partitioning in 3-MGA/AS droplets*

In addition to measurements and thermodynamic phase equilibrium calculations, we carried out MD simulations to probe the distribution of the species within 3-MGA/AS droplets at different RH (for details see MD simulations, Supporting material). In the MD simulations, 3-MGA molecules tended to stay near the surface while the $NH_4^+$ and $SO_4^{2-}$ ions preferred to stay in the droplet bulk. In a previous study, Gopalakrishnan et al. (2005) carried out polarizable MD simulations to study the propensity of $NH_4^+$ and $SO_4^{2-}$ for the air–liquid interface. They showed that $NH_4^+$ ions have a stronger preference for the interface than $SO_4^{2-}$. With the addition of 3-MGA near the water interface in our MD simulations, we have observed similar results (see Figures S11 and S12 in the supplemental information), namely that $NH_4^+$

prefers proximity to the interface more so than $SO_4^{2-}$. The presence of 3-MGA may have
pushed the maximum densities of $NH_4^+$ and $SO_4^{2-}$ slightly more towards the bulk relative to
the interface, but such differences may have been due to the differences in system sizes and
other simulation parameters. From the determined density profiles, the 3-MGA molecules are
concentrated near the surface in both droplet and slab geometry setups. Even in the most dilute
case, which responds to RH > 90%, we did not observe an entirely homogeneous liquid phase
as perceived from the experiments. With the highest number of water molecules (corresponding
to the highest RH environment) in our simulations, a few 3-MGA molecules were able to
diffuse into the bulk while the majority remained located near the surface, suggesting a
(delayed) start of coming to a single phase. A possible cause for the discrepancy is the
underestimated magnitude of molecular interactions between water and 3-MGA in our
simulations. However, considering the detailed molecular-level resolution by which MD
simulations allow us to probe surface and near-surface inhomogeneities in composition, this
method provides an additional perspective on the bulk–surface partitioning of 3-MGA with or
without LLPS occurring. In this context, equilibrium thermodynamics, when accounting for
the surface area to volume ratio of small droplets, hence non-negligible interfacial energies,
would also hint at expected inhomogeneities at/near surfaces. We note that a considerable
amount of effort is required to further optimize the molecular models, e.g. with more elaborate
search for better parameters, functional forms and/or more sophisticated models. Since this is
not the primary focus of this study, we have decided to defer such further simulations to future
work.

**Conclusions and Atmospheric Implications**
In this work, we investigated the heterogeneous kinetics of aqueous 3-MGA/AS droplets with
an OIR of 1. LLPS onset by spinodal decomposition is revealed to occur at 74.6 % RH (by
optical microscopy) and ~72 % (by LQ-EDB). Partially engulfed, core–shell, and transition
from core–shell to partially engulfed configurations are observed. Kinetic data show that
phase-separated droplets have a slightly higher reactivity towards gas-phase OH radicals,
compared to single-phase aqueous droplets (measured at higher RH). As phase separation
occurs in this system at lower RH than the single-phase state, an uneven distribution of 3-MGA
within the droplets resulted in the former case, increasing the collision probability between 3-
MGA molecules and OH radicals at or near the droplet surface, supported by the shorter
average distance between the OH radical and its nearest organic molecules in phase-separated
droplets. The change in reactivity due to the presence or absence of phase separation may affect

the atmospheric lifetimes of organic compounds against heterogeneous oxidation. For both single-phase and phase-separated aerosols, the heterogeneous reactivity of 3-MGA/AS droplets shows an increasing trend as the RH becomes lower. It could be attributed to a higher surface concentration of 3-MGA at lower RH which intensifies the parent organic molecule decay initiated by OH radical and the secondary chemical reactions, thus enhancing the overall heterogeneous reactivity and leading to shorter atmospheric lifetimes. For instance, using the kinetic data and a 24-h averaged gas-phase OH concentration of $1.5 \times 10^6$ molecules $cm^{-3}$, the lifetime of 3-MGA against heterogeneous OH oxidation is estimated to decrease from $7.01 \pm 0.13$ days to $4.46 \pm 0.05$ days when RH decreases from 88% to 55%. Together with previously published results for aqueous 3-MGA/AS droplets consisting of a single liquid phase, the heterogeneous reactivity is positively correlated with the initial mass fraction of 3-MGA (**Figure S6**), suggesting that the bulk concentration of organic molecules could be chosen as a proxy to parameterize the heterogeneous reactivity.

We have performed thermodynamic phase equilibrium calculations to understand the phase transition and composition of 3-MGA/AS aerosols. The AIOMFAC-LLE model simulations provide an explanation for the phase separation mechanism consistent with the observations. While the model slightly overpredicts the SRH compared to the measurements, further development of predictive group contribution models like AIOMFAC-LLE would be highly desirable to better predict phase transitions and other composition-dependent properties of the typically highly complex organic–inorganic mixtures representative of atmospheric aerosols. In addition to thermodynamic modeling, our MD simulations suggest that 3-MGA molecules have a propensity to partition to the near-surface layer of a droplet, even at high RH. This is in qualitative support of the finding that the 3-MGA reactivity between single-phase and LLPS states of the droplets might not be dramatically different. Further improvements of MD simulation details for the studied system will likely lead to a valuable, complementary tool, because of the available molecular details it can provide. The combination of MD simulations and equilibrium thermodynamic computations will allow us to better understand the phase separation, morphology and size properties of aerosols, which ultimately govern the heterogeneous reactivity and other atmospheric processes, e.g. dynamic gas–particle partitioning.

Laboratory studies have shown that an organic-rich outer shell is always formed for hydrophobic organic compounds in a LLPS scenario, possibly shielding the interior from

surface reactions with gas-phase oxidants (McNeill et al., 2007; Li et al., 2020). The organic-rich outer shell (Arangio et al., 2015; Houle et al., 2018) could be viscous and the heterogeneous reactivity could be limited by the diffusion of organic species or oxidants across the organic outer shell. On the other hand, our results show that aqueous organic–inorganic droplets with more hydrophilic organic compounds (e.g. 3-MGA) may not necessarily experience diffusion limitation during heterogeneous OH oxidation, even when phase-separated. The overall heterogeneous reactivity is likely governed by the surface concentration of organic molecules at room temperature. It acknowledges that when the temperature decreases, the aerosol viscosity generally increases (everything else being equal). This would lead to a decrease in the diffusion rate of species from the bulk to the surface where oxidation preferentially takes place, and the overall rate of the oxidation will become more likely controlled by the diffusion. This is an expected temperature effect in the boundary layer (e.g. in the cold season or cold climates). However, in the context of vertical air motions (e.g. when air parcels rise adiabatically), a decrease in temperature will be accompanied by changes in RH; in the case of adiabatic ascent RH tends to increase. This in turn would potentially limit the increase in viscosity of hygroscopic aerosols or even lower it while RH remains high (Gervasi et al., 2020). Overall, this work further emphasizes that the effects of phase separation and potentially distinct aerosol morphologies add further complexity to the quantitative understanding of the heterogeneous reactivity of organic compounds in aqueous organic–inorganic droplets in the atmosphere, motivating further experimental and process modeling studies for a variety of aerosol systems.

**Data availability**

Data are available upon request from the corresponding author.

**Author contributions**

Hoi Ki Lam and Man Nin Chan designed and ran the experiments. Jack Choczynski, James F. Davies, Dongwan Ham and Mijung Song provided the phase separation data. Andreas Zuend, Wentao Li, Ying-Lung Steve Tse provided the model simulations. Hoi Ki Lam, Rongshuang Xu and Man Nin Chan prepared the manuscript. All co-authors provided comments and suggestions to the manuscript.

**Competing interests**

The authors declare that they have no conflict of interest.

**Acknowledgements**

This work is supported by the Hong Kong Research Grants Council (HKRGC) Project ID: 2191111 (Ref 24300516). Andreas Zuend acknowledges support by the Natural Sciences and Engineering Research Council of Canada (NSERC) (grant no. RGPIN/04315-2014). Mijung Song acknowledges support by the National Research Foundation of Korea grant funded by the Korea Government (NRF-2019R1A2C1086187). James F. Davies acknowledges support of UC Riverside through startup funding

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

**Table 1.** Chemical structure, effective heterogeneous OH rate constant and effective uptake coefficient of aqueous droplets containing 3-MGA and ammonium sulfate (AS) in an organic-to-inorganic dry mass ratio (OIR) = 1 at different RH at 293 K.

| Chemical structure |  | | | | | | | |
|---|---|---|---|---|---|---|---|---|
| Chemical formula | $C_6H_{10}O_4$ | | | | | | | |
| Separation RH (SRH) (%RH) | 74.6 (optical microscopy) <br> 72 (LQ-EDB) | | | | | | | |
| Morphology of Phase Separated Droplets | Core shell morphology, partially engulfed morphology and <br> a transition from core shell morphology to partially engulfed morphology | | | | | | | |
| RH (%) | 55 | 60 | 65 | 70 | 75 | 80 | 85 | 88 |
| $k$ | 1.73 ± 0.02 | 1.65 ± 0.02 | 1.61 ± 0.03 | 1.34 ± 0.03 | 1.28 ± 0.01 | 1.19 ± 0.02 | 1.06 ± 0.01 | 1.01 ± 0.02 |
| $\gamma_{eff}$ | / | / | / | / | 0.40 ± 0.03 | 0.33 ± 0.02 | 0.25 ± 0.02 | 0.20 ± 0.01 |

Note: $k$ is the effective heterogeneous OH rate constant ($10^{-12}$ cm$^3$ molecule$^{-1}$ s$^{-1}$); $\gamma_{eff}$ = effective OH uptake coefficient

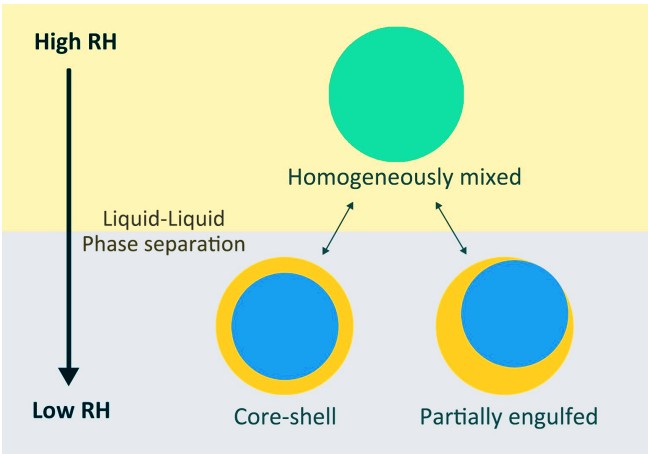

**Figure 1.** A simplified diagram for the phase separation in aqueous droplets containing inorganic salts, organic compounds, and water. At high humidity, aqueous droplets exist as homogeneously mixed liquid. When the humidity decreases, they undergo phase separation, leading to different morphologies such as core–shell and partially engulfed structures. Blue indicates the aqueous inorganic-rich phase; Yellow indicates the organic-rich phase; Bluish green represents the homogeneously mixed single liquid phase case.

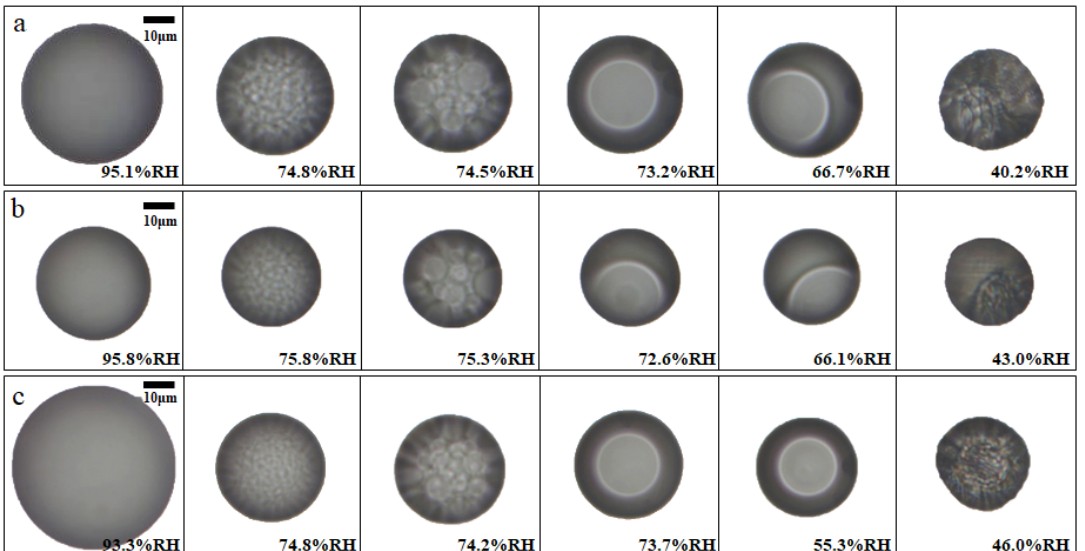

**Figure 2.** Image sequences of liquid–liquid phase separation and efflorescence leading to (a) the morphological transition from core–shell to partially engulfed, (b) partially engulfed morphology and (c) core–shell morphology of 3-MGA/AS droplets (OIR = 1) at 290 ± 1 K upon dehumidification at a rate of 0.2-0.3 % RH min$^{-1}$. The size bar in the figure corresponds to 10 μm.

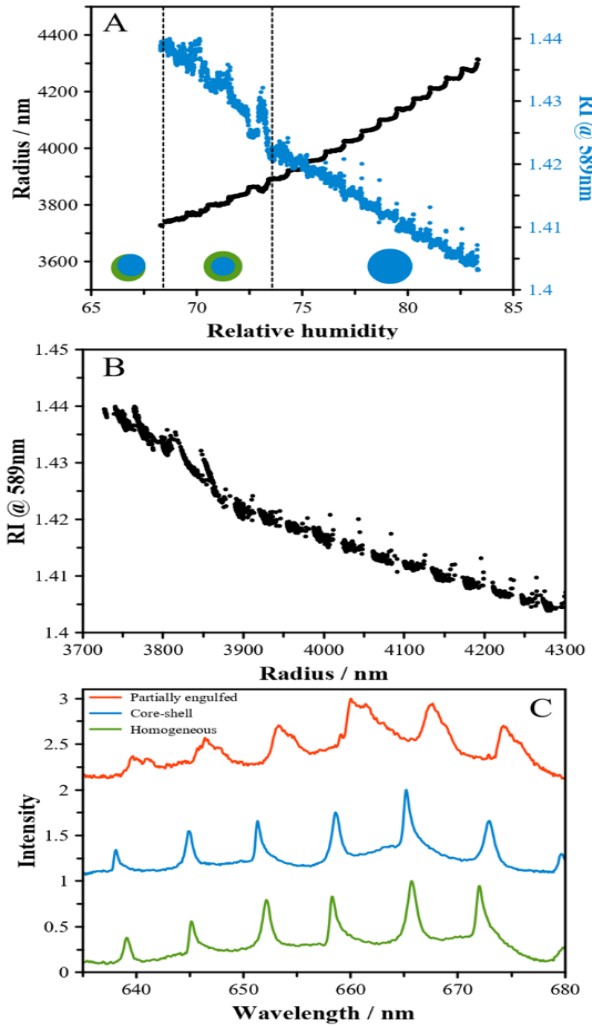

**Figure 3.** (A) Radius (black) and refractive index (blue) of an aqueous 3-MGA/AS droplet (OIR = 1) levitated in a LQ-EDB as a function of RH upon dehumidification at a temperature of 298K; (B) The RI as a function of radius as the RH is decreased. LLPS is identified as a discontinuous change in the slope of this data; (C) Mie resonance spectra of an aqueous 3-MGA/AS droplet levitated in a LQ-EDB recorded prior to LLPS (green), following LLPS to form a core–shell morphology (blue) and following formation of a partially engulfed morphology (red).

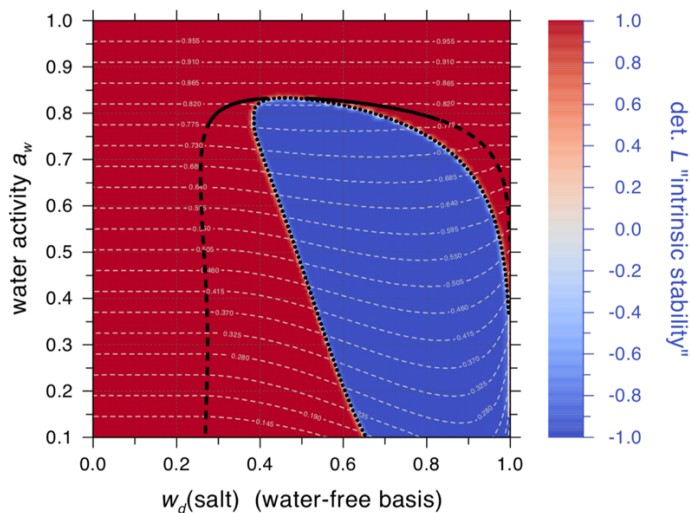

**Figure 4.** Predicted equilibrium phase diagram of the aqueous 3-MGA/AS system as a function of RH (or water activity, $a_w$ = RH/100 %) and mass fraction of AS in the dry 3-MGA/AS mixture, $w_d$(salt). Predictions by AIOMFAC-LLE for $T$ = 293 K with crystallization of AS suppressed. Bold black solid and dashed curves: the binodal limit of single-phase stability, indicating a single liquid phase to be stable "outside" of it (at higher $a_w$ or low salt content). The dashed branches of the binodal curve indicate that dissolved AS in solution would be supersaturated relative to crystalline AS at those compositions. Dotted curve: spinodal limit of stability, with a single liquid phase being intrinsically unstable inside the area bounded by it (blue area). Red area: the composition space where a single liquid phase is stable or metastable with regard to LLPS (color scale has arbitrary units). Thin, grey dashed lines: equilibrium water activity contours (tie-lines within LLE region). An OIR = 1 corresponds to $w_d$(salt) = 0.5, which leads to a near-immediate crossing of the spinodal limit when such a mixture is dried from high water activity (while $w_d$(salt) = 0.5 remains constant).

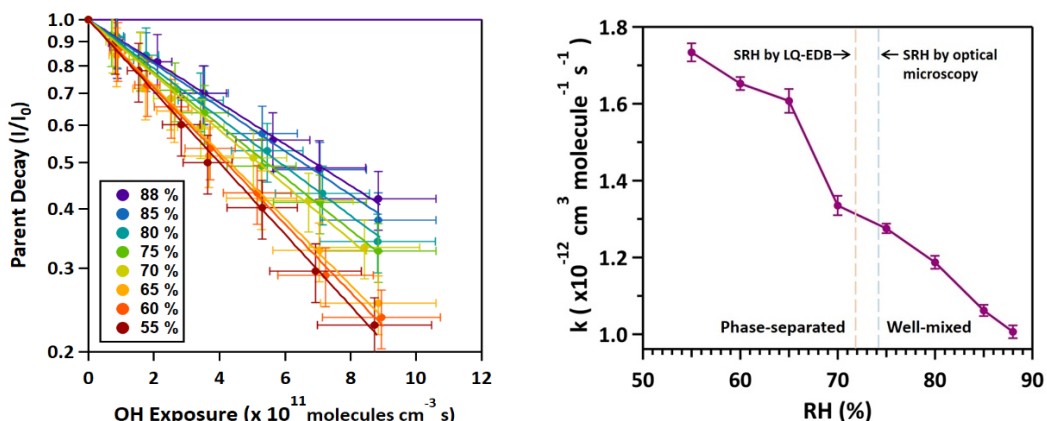

**Figure 5.** (**Left**) Normalized parent decay of 3-MGA/AS aerosols (OIR = 1) as a function of RH; (**Right**) The effective second-order heterogeneous OH reaction rate constant (*k*) plotted against the relative humidity (RH) for the heterogeneous OH oxidation of 3-MGA/AS aerosols (OIR = 1).