# Peer review of "Effects of Liquid-Liquid Phase Separation and Relative Humidity on the Heterogeneous OH Oxidation of Inorganic-Organic Aerosols: Insights from Methylglutaric Acid/Ammonium Sulfate Particles"

_Atmospheric Chemistry and Physics, 2020_

## Referee Comment (RC1) · Anonymous Referee #1 · 10 Oct 2020

Here follows the review of the manuscript entitled "Effects of Liquid–Liquid Phase Separation and Relative Humidity on the Heterogeneous OH Oxidation of Inorganic-Organic Aerosols: Insights from Methylglutaric Acid/Ammonium Sulfate Particles" by Lam et al. In this laboratory work the authors study how the OH heterogenous reactivity changes as inorganic/organic particles composed of ammonium sulfate (AS) and 3-methyglutaric acid (MGA) undergo liquid-liquid phase separation (LLPS) in response to humidity changes. This work is, in part, based on the previously applied experi-

mental procedures by Xu et al. (ACP, 2020). Particles with an organic-to-inorganic dry mass ratio (OIR) of 1 experience LLPS at about 75% RH displaying core-shell or partially engulfed morphology. The authors observe that OH reactivity is higher for the LLPS state compared to the purely liquid phase state. Application of phase separation measurements and model simulations suggest that the enhanced reactivity is due to the higher concentration of MGA at the droplet surface which in turn increases the reactive collision probability between MGA and OH. Model simulations of the diffusion of MGA, including the core-shell configuration, provide reasoning that the diffusivity of MGA is sufficiently fast that reactivity is not diffusion limited. Therefore, the authors conclude that MGA surface concentration is likely the determining factor of the overall observed OH reactivity.

The topic of this study fits very well within the scope of Atmospheric Chemistry and Physics. I enjoyed reading this manuscript. The experimental approach and methods appear to be sound and present a continuation and extension of a previously published study by this group. This work adds significantly to our understanding of how particle phase changes impact gas-to-particle interactions, specifically the chemical oxidation of organic particular matter. I only suggest minor revisions.

For the molecular dynamics simulations, it would be nice to set those results in context to previous work. Do the general results/trends observed here agree with previous studies? For example, the work by Tobias and Jungwirth groups examining the distribution of ions in the aqueous phase. One would assume $SO4^{2-}$ being more in the bulk than $NH4^+$. However, in the presence of an organic surfactant this may change. Some discussion referring to previous MD studies on aqueous solutions and presence of surfactants should be added.

As stated AIOMFAC-LLE (VISC) is a group-contribution model and not specifically setup to simulate the organic–inorganic system studied here. However, when looking at the results displayed in Fig. 4, is there a way to give the range of uncertainties in shown values derived by this model? I assume the model is fit to observational data

of single component data sets, etc. What would be the expected value ranges for the binodal limits, water activity, etc.? This may not be easy to answer but a best-guess of value ranges would be appreciated. Also, I believe "LLE" is not spelled out.

Line 58: The study by Slade et al. (2015) and (2017) could be added here which relate OH uptake with particle hygroscopicity of amorphous organic and inorganic/organic particles.

Line 61: The authors might add the recent study by Li et al. (2020) on OH uptake by organic matter in various phase states.

Line 72-78: Potentially relevant to this study: Charnawskas et al. (2017) documented core-shell morphology of submicron inorganic/organic particles using X-ray microscopy (similar to this study, i.e., core-shell).

Line 292: I feel this sentence is missing a word. The single liquid phase has an order of magnitude...greater than what? Maybe I misunderstand this sentence.

Line 450: What are the potential uncertainties in AIOMFAC-VISC and thus the uncertainties in the time scale for diffusive mixing? Since the values are close to the time of collision events, it may be good to have a boundary on those theoretically derived values.

Line 486: See my comments above on MD studies.

Line 552-555: The study by Li et al. (2020) may be relevant for this statement.

Technical correction:

Line 373: I suggest to omit "occurred".

References:

Slade, J. H., Thalman, R., Wang, J., and Knopf, D. A.: Chemical aging of single and multicomponent biomass burning aerosol surrogate particles by OH: implications

for cloud condensation nucleus activity, Atmos. Chem. Phys., 15, 10183–10201, doi:10.5194/acp-15-10183-2015, 2015.

Slade, J. H., Shiraiwa, M., Arangio, A., Su, H., Pöschl, U., Wang, J., and Knopf, D. A.: Cloud droplet activation through oxidation of organic aerosol influenced by temperature and particle phase state, Geophys. Res. Lett., 44, 1583–1591, https://doi.org/10.1002/2016gl072424, 2017.

Li, J., Forrester, S. M., and Knopf, D. A.: Heterogeneous oxidation of amorphous organic aerosol surrogates by O3, NO3, and OH at typical tropospheric temperatures, Atmos. Chem. Phys., 20, 6055–6080, https://doi.org/10.5194/acp-20-6055-2020, 2020.

Charnawskas, J. C., Alpert, P. A., Lambe, A. T., Berkemeier, T., O'Brien, R. E., Massoli, P., Onasch, T. B., Shiraiwa, M., Moffet, R. C., Gilles, M. K., Davidovits, P., Worsnop, D. R., and Knopf, D. A.: Condensed-phase biogenic-anthropogenic interactions with implications for cold cloud formation, Faraday Discuss., 200, 165–194, https://doi.org/10.1039/c7fd00010c, 2017.
* * *

---

## Referee Comment (RC2) · Anonymous Referee #2 · 26 Oct 2020

General comments. The manuscript presents results from studies probing the effects of liquid-liquid phase separations on the loss rate for methylglutaric acid signal through heterogeneous OH oxidation. A range of different analyses were combined with the flow tube studies to fully characterize the system including optical microscopy, an electrodynamic balance, and modeling studies. The authors found that the heterogeneous OH oxidation rate increased in LLPS particles, likely due to increased organic concentrations near the surface in the particles. Overall the paper is well written and the

conclusions are supported by the data. There are a few places where additional information would enable a broader view of the results. I recommend this manuscript for publication in ACP after the following minor comments are addressed.

Specific Comments

1. The effective heterogeneous OH rate constant was reported to vary from 1.01 x 10ˆ-12 to 1.73 x 10ˆ-12 cmˆ3 moleculeˆ-1 sˆ-1. How does this scale to lifetimes in the atmosphere? How much of a difference might be expected for the lifetimes of organic compounds in LLPS systems in the atmosphere?

2. Where do the various error estimates come from? Are these from fits or from replicate measurements (or both)?

3. In the discussion of diffusivity, the comparison is made for laboratory studies. How would this extrapolate to temperatures found in the atmosphere? Could we still anticipate that diffusion would not be limiting, especially given the lower OH radical concentrations?

4. The kinetics were tracked by looking at the loss of the parent signal, and the same products appear to be formed in the experiments. However, the intensities of these products have some apparent differences in Figure S4. Was there any correlation of product ion signals to the decay rate of the parent ion? Either in terms of the relative intensities between C6H9O5- or C6H7O5- or the total product ion signal?

5. Figure 1 is not interpretable in black and white, I suggest a different color scheme, or more gradation.

6. What are the error bars on Figure 5 a (how are they estimated)? Are there error bars that can be applied to Figure 5b?

---

## Author Comment (AC1) · 20 Dec 2020

Here follows the review of the manuscript entitled "Effects of Liquid–Liquid Phase Separation and Relative Humidity on the Heterogeneous OH Oxidation of Inorganic-Organic Aerosols: Insights from Methylglutaric Acid/Ammonium Sulfate Particles" by Lam et al. In this laboratory work the authors study how the OH heterogenous reactivity changes as inorganic/organic particles composed of ammonium sulfate (AS) and 3- methyglutaric acid (MGA) undergo liquid-liquid phase separation (LLPS) in response to humidity changes. This work is, in part, based on the previously applied experimental procedures by Xu et al. (ACP, 2020). Particles with an organic-to-inorganic dry mass ratio (OIR) of 1 experience LLPS at about 75% RH displaying core-shell or partially engulfed morphology. The authors observe that OH reactivity is higher for the LLPS state compared to the purely liquid phase state. Application of phase separation measurements and model simulations suggest that the enhanced reactivity is due to the higher concentration of MGA at the droplet surface which in turn increases the reactive collision probability between MGA and OH. Model simulations of the diffusion of MGA, including the core-shell configuration, provide reasoning that the diffusivity of MGA is sufficiently fast that reactivity is not diffusion limited. Therefore, the authors conclude that MGA surface concentration is likely the determining factor of the overall observed OH reactivity.

The topic of this study fits very well within the scope of Atmospheric Chemistry and Physics. I enjoyed reading this manuscript. The experimental approach and methods appear to be sound and present a continuation and extension of a previously published study by this group. This work adds significantly to our understanding of how particle phase changes impact gas-to-particle interactions, specifically the chemical oxidation of organic particular matter. I only suggest minor revisions.

**We would like to sincerely thank the reviewer for his/her thoughtful comments and suggestions. Please see our responses to reviewer's comments and suggestions below.**

**Reviewer's Comment #1**
For the molecular dynamics simulations, it would be nice to set those results in context to previous work. Do the general results/trends observed here agree with previous studies? For example, the work by Tobias and Jungwirth groups examining the distribution of ions in the aqueous phase. One would assume $SO_4^{2-}$ being more in the bulk than $NH_4^+$. However, in the presence of an organic surfactant this may change. Some discussion referring to previous MD studies on aqueous solutions and presence of surfactants should be added.

**Authors' Response**
We would like to thank the reviewer for suggesting a comparison with the previous results. We put the density profiles of our $SO_4^{2-}$ and $NH_4^+$ models in Figures S11 (slabs) and S12 (droplets) in the supplement information. Our models are consistent with the previous study by Gopalakrishnan et al. (2005) that shows $NH_4^+$ prefers the interface more so than $SO_4^{2-}$. In our study, the presence of 3-MGA does not change this trend. 3-MGA may have pushed the maximum densities of $NH_4^+$ and $SO_4^{2-}$ slightly more towards the bulk when compared to the previous study, but the differences in the system sizes and simulation parameters may have played a role.

We have now added the following the main text in the revised manuscript.

Line 499, "In a previous study, Gopalakrishnan et al. (2005) carried out polarizable MD simulations to study the propensity of $NH_4^+$ and $SO_4^{2-}$ for the air–liquid interface. They showed that $NH_4^+$ ions have a stronger preference for the interface than $SO_4^{2-}$. With the addition of 3-MGA near the water interface in our MD simulations, we have observed similar results (see Figures S11 and S12 in the supplemental information), namely that $NH_4^+$ prefers proximity to the interface more so than $SO_4^{2-}$. The presence of 3-MGA may have pushed the maximum densities of $NH_4^+$ and $SO_4^{2-}$ slightly more towards the bulk relative to the interface, but such differences may have been due to the differences in system sizes and other simulation parameters."

Reference
Gopalakrishnan, S.; Jungwirth, P.; Tobias, D. J.; Allen, H. C. Air−Liquid Interfaces of Aqueous Solutions Containing Ammonium and Sulfate: Spectroscopic and Molecular Dynamics Studies. J. Phys. Chem. B, 109, 8861–8872, 2005.

**Reviewer's Comment #2**
As stated AIOMFAC-LLE (VISC) is a group-contribution model and not specifically setup to simulate the organic–inorganic system studied here. However, when looking at the results displayed in Fig. 4, is there a way to give the range of uncertainties in shown values derived by this model? I assume the model is fit to observational data of single component data sets, etc. What would be the expected value ranges for the binodal limits, water activity, etc.? This may not be easy to answer but a best-guess of value ranges would be appreciated. Also, I believe "LLE" is not spelled out.

**Authors' Response**
Thanks for the comment. This question and our answer refer to Figure 4 and associated text in Sect. 3.2. The reviewer is correct in stating that providing a range of uncertainties for the binodal limit curve of the liquid–liquid equilibrium (LLE) prediction is not easy, as will be explained in the following. Absent reliable quantitative measurements of the binodal curve for this system, we can only provide estimates supported by limited evidence. We will also spell out the abbreviations LLE and AIOMFAC in the revised manuscript.

The predicted phase diagram shown in Figure 4 for the ternary aqueous 3-MGA/AS system with OIR = 1 has been computed by using the previously determined parameter set of the AIOMFAC group-contribution model from the work by Zuend et al. (2011). A specific dataset of this ternary system was not directly involved in the optimization of the AIOMFAC model parameters and, hence, the model is not expected to perform optimally for this system. However, the training and optimization of the AIOMFAC model by Zuend et al. (2011) involved one multicomponent data set of three dicarboxylic acids with 6 carbon atoms, including 3-methylglutaric acid, as well as ammonium sulfate (Fig. S0220 from the supplementary information document of Zuend et al. 2011; reproduced below for reference). That system is expected to behave similarly to the ternary 3-MGA/AS system from our current study, although the involved OIR differ.

[Figure]

Fig. S0220 (AIOMFAC_output_1059)

H$_2$O (1) + 2-Methylglutaric_acid (2) + 3-Methylglutaric_acid (3) + 2,2-Dimethylsuccinic_acid (4) + (NH$_4$)$_2$SO$_4$ (5)

Temperature range: 291 -- 293 K

**Fig. S0220** reproduced from Zuend et al. (2011); shown in their supplementary information. Model–measurement comparison for the system water (1) + 2-Methylglutaric acid (2) + 3-Methylglutaric acid (3) + 2,2-Dimethylsuccinic acid (4) + ammonium sulfate (5) at temperatures near 293 K. The mixing ratio among the organic diacids is 1:1:1 by mass. Composition is given in mole fractions ($x$). Cross symbols mark water activity measurements by an electrodynamic balance or a water activity meter at higher $a_w$, open circles and error bars are the model predictions by AIOMFAC pertaining to each composition point. The error bars indicate cumulative AIOMFAC prediction sensitivity to a composition uncertainty of 0.01 in mole fraction; see details in Zuend et al. (2011).

Fig. S0220 indicates that measured and predicted water activities are approximately in agreement when accounting for a mole fraction composition uncertainty of about 0.01, which can lead to larger uncertainty in predicted water activity (error bars in the attached figure) of approximately ±5 % at water activities above ~0.6. Collectively, the model predictions also show a slight high bias in predicted water activity compared to the measurements, which may explain at least partially why the onset of liquid–liquid phase separation predicted by AIOMFAC-LLE in Figure 4 for OIR = 1 is at a higher water activity of about 0.83 than the one determined by the microscopy experiments ($a_w$ ~0.75).

Therefore, we estimate that the onset RH of LLPS is predicted with a potential high bias of about 5–8%. From this work and previous comparisons, e.g. Song et al. (2012a) it seems to be the case that for systems involving dicarboxylic acids and ammonium sulfate AIOMFAC tends to predict a higher onset RH of LLPS than is usually determined from droplet or bulk measurements. This is mentioned in Sect. 3.2, lines 271–274 in the original manuscript. Estimating the error in the extent (or area) of phase separation indicated by the "width" of the

area enclosed by the binodal curve in Figure 4 in terms of the mass fraction of salt on water-free basis, is difficult. Given that the Gibbs energy difference between a single mixed phase and LLPS is relatively small in the composition space near the binodal limit (and between binodal and spinodal curves), uncertainties in the predicted activity coefficients of the mixture components can amplify. To provide a rough estimate, we would also expect about 0.1 units of dry mass fraction uncertainty in $w_d$(salt). The location of the critical point (where binodal and spinodal curves touch) is expected to remain in the range of $w_d$(salt) = 0.4 to 0.5.

We have made the following changes to the manuscript.

We rephrase the sentence to define abbreviations. Line 262: "Thermodynamic phase equilibrium calculations were also performed using the Aerosol Inorganic–Organic Mixtures Functional groups Activity Coefficients (AIOMFAC) liquid–liquid equilibrium (LLE) model, hereafter referred to as AIOMFAC-LLE model, to compare the results of the experimentally observed LLPS range and onset mechanism (Zuend et al., 2008, 2010, 2011; Zuend and Seinfeld, 2013)."

Line 276: we add "Based on this comparison and other related comparisons of AIOMFAC-LLE predictions with measurements (Song et al., 2012a), we estimate that the onset of LLPS is predicted within about 10 % uncertainty in RH."

**Reviewer's Comment #3**
Line 58: The study by Slade et al. (2015) and (2017) could be added here which relate OH uptake with particle hygroscopicity of amorphous organic and inorganic/organic particles.

**Authors' Response**
We have added these two references.

**Reviewer's Comment #4**
Line 61: The authors might add the recent study by Li et al. (2020) on OH uptake by organic matter in various phase states.

**Authors' Response**
We have added this reference.

**Reviewer's Comment #5**
Line 72-78: Potentially relevant to this study: Charnawskas et al. (2017) documented core-shell morphology of submicron inorganic/organic particles using X-ray microscopy (similar to this study, i.e., core-shell).

**Authors' Response**
We have added this reference.

**Reviewer's Comment #6**
Line 292: I feel this sentence is missing a word. The single liquid phase has an order of magnitude. . .greater than what? Maybe I misunderstand this sentence.

**Authors' Response**
Thanks for the comment. In the original manuscript, we intended to mention that the viscosity of an aqueous 3-MGA/AS droplet with a single liquid phase ranges from $\sim 10^{-3}$ Pa s to $10^{-2}$ Pa s. We have revised the sentence in the revised manuscript.

Line 296, "As shown in Figure S3, when the RH > SRH, the viscosity of an aqueous 3-MGA/AS droplet with a single liquid phase ranges from $\sim 10^{-3}$ to $10^{-2}$ Pa s and increases with decreasing RH."

**Reviewer's Comment #7**
Line 450: What are the potential uncertainties in AIOMFAC-VISC and thus the uncertainties in the time scale for diffusive mixing? Since the values are close to the time of collision events, it may be good to have a boundary on those theoretically derived values.

**Authors' Response**
The uncertainty in predicted viscosities of the organic-rich phase is indicated in Fig. S3 by the red shaded area. These uncertainty estimates were generated using AIOMFAC-VISC by accounting for a $\pm 5$ % uncertainty in the estimated glass transition temperatures of the pure components. Here, the uncertainties are about -0.21 to +0.34 in $\log_{10}$[viscosity / (Pa s)] units for the RH range from 55 % to 70 %. To provide additional data on viscosity, diffusivity and mixing timescale estimates, we have added additional data in Table S1 for these properties of the organic-rich phase. Aside from the already listed best estimates, we list now also the estimated lower and upper bounds for these predicted parameters. For example, the uncertainty in the predicted phase viscosity translates to estimated bounds on the mixing timescale of 3.6 to 13.8 µs at 60 % RH; with a best estimate of 6.6 µs.

We list now also predicted lower and upper bounds for values related to the viscosity predictions in Table S1.

**Excerpt of revised Table S1** showing organic-rich phase viscosities (AIOMFAC-VISC estimate) at 293 K and associated diffusion coefficients and mixing times. Lower and upper value estimates for phase viscosities and derived diffusivity values are listed in brackets (based on 5 % uncertainty in pure-component glass transition temperature).

| RH (%) | 55 | 60 | 65 | 70 | 75 | 80 | 85 | 88 |
|---|---|---|---|---|---|---|---|---|
| Viscosity of the organic-rich phase (Pa s) | 0.0958 [0.051; 0.21] | 0.0636 [0.035; 0.133] | 0.0421 [0.024; 0.083] | 0.0275 [0.016; 0.051] | / | / | / | / |
| Diffusion coefficient of 3-MGA molecules in organic-rich phase ($\times 10^{-12}$ m$^2$ s$^{-1}$) | 6.15 [11.6; 2.79] | 9.26 [16.8; 4.43] | 14.0 [24.2; 7.09] | 21.4 [35.3; 11.5] | / | / | / | / |
| Diffusive mixing time of organic-rich phase (µs) | 10.2 [5.4; 22.5] | 6.6 [3.6; 13.8] | 4.2 [2.4; 8.4] | 2.6 [1.6; 4.9] | / | / | / | / |

**Reviewer's Comment #8**
Line 486: See my comments above on MD studies.

**Authors' Response**
Please kindly see our response to Reviewer 1's Comment #2.

**Reviewer's Comment #9**
Line 552-555: The study by Li et al. (2020) may be relevant for this statement.

**Authors' Response**
We have added this reference.

**Reviewer's Comment #10**
Technical correction: Line 373: I suggest to omit "occurred".

**Authors' Response**
We have made the correction.

References

Slade, J. H., Thalman, R., Wang, J., and Knopf, D. A.: Chemical aging of single and multicomponent biomass burning aerosol surrogate particles by OH: implications for cloud condensation nucleus activity, Atmos. Chem. Phys., 15, 10183–10201, 2015.

Slade, J. H., Shiraiwa, M., Arangio, A., Su, H., Pöschl, U., Wang, J., and Knopf, D. A.: Cloud droplet activation through oxidation of organic aerosol influenced by temperature and particle phase state, Geophys. Res. Lett., 44, 1583–1591, 2017.

Li, J., Forrester, S. M., and Knopf, D. A.: Heterogeneous oxidation of amorphous or- ganic aerosol surrogates by $O_3$, $NO_3$, and OH at typical tropospheric temperatures, At- mos. Chem. Phys., 20, 6055–6080, 2020.

Charnawskas, J. C., Alpert, P. A., Lambe, A. T., Berkemeier, T., O'Brien, R. E., Mas- soli, P., Onasch, T. B., Shiraiwa, M., Moffet, R. C., Gilles, M. K., Davidovits, P., Worsnop, D. R., and Knopf, D. A.: Condensed-phase biogenic-anthropogenic inter- actions with implications for cold cloud formation, Faraday Discuss., 200, 165–194, 2017.

---

## Author Comment (AC2) · 20 Dec 2020

General comments. The manuscript presents results from studies probing the effects of liquid-liquid phase separations on the loss rate for methylglutaric acid signal through heterogeneous OH oxidation. A range of different analyses were combined with the flow tube studies to fully characterize the system including optical microscopy, an electrodynamic balance, and modeling studies. The authors found that the heterogeneous OH oxidation rate increased in LLPS particles, likely due to increased organic concentrations near the surface in the particles. Overall the paper is well written and the conclusions are supported by the data. There are a few places where additional information would enable a broader view of the results. I recommend this manuscript for publication in ACP after the following minor comments are addressed.

**We would like to sincerely thank the reviewer for his/her thoughtful comments and suggestions. Please see our responses to reviewer's comments and suggestions below.**

Specific Comments

**Reviewer's Comment #1**
The effective heterogeneous OH rate constant was reported to vary from 1.01 x 10^-12 to 1.73 x 10^-12 cm^3 molecule^-1 s^-1. How does this scale to lifetimes in the atmosphere? How much of a difference might be expected for the lifetimes of organic compounds in LLPS systems in the atmosphere?

**Authors' Response**
Thanks for the comment. Using a 24-h averaged gas-phase OH concentration of $1.5 \times 10^6$ molecules cm$^{-3}$, the lifetime of 3-MGA against heterogeneous OH oxidation is estimated to decrease from $7.01 \pm 0.13$ days to $4.46 \pm 0.05$ days when the effective heterogeneous OH rate constant increases from $1.01 \pm 0.02 \times 10^{-12}$ to $1.73 \pm 0.02 \times 10^{-12}$ cm$^3$ molecule$^{-1}$ s$^{-1}$. These results would suggest that the lifetime of 3-MGA in phase-separated droplets would be shorter compared to that in single-phase aqueous droplets in the atmosphere. We also acknowledge that the presence of other organic and inorganic components in atmospheric aerosols is expected to further affect this estimated lifetime (e.g. if additional low-polarity organic components in the shell phase or air–liquid interface replace some of the 3-MGA exposed to the gas phase, the lifetime may prolong again).

We have added this information in the revised manuscript.
Conclusion, Line 543, "For instance, using the kinetic data and a 24-h averaged gas-phase OH concentration of $1.5 \times 10^6$ molecules cm$^{-3}$, the lifetime of 3-MGA against heterogeneous OH oxidation is estimated to decrease from $7.01 \pm 0.13$ days to $4.46 \pm 0.05$ days when RH decreases from 88% to 55%."

**Reviewer's Comment #2**
Where do the various error estimates come from? Are these from fits or from replicate measurements (or both)?

**Authors' Response**
The errors of effective OH uptake coefficient, $\gamma_{eff}$ were determined according to the error propagation rule (Guidelines for Evaluating and Expressing the Uncertainty of NIST Measurement Results: http://physics.nist.gov/TN1297):

$$\sigma_\gamma = \gamma_{eff} * \sqrt{\left(\frac{\sigma_k}{k}\right)^2 + \left(\frac{\sigma_{D0}}{D_0}\right)^2 + \left(\frac{\sigma_{\rho_0}}{\rho_0}\right)^2 + \left(\frac{\sigma_{mf}}{mf}\right)^2} \qquad (1)$$

where $\sigma_\gamma$ is the error of effective OH uptake coefficient, $\gamma_{eff}$ is the best estimate of the effective OH uptake coefficient, $k$ is the fitted effective heterogeneous OH rate constant, $\sigma_k$ is the uncertainty of effective heterogeneous OH rate constant, $D_0$ is the mean surface-weighted diameter, $\sigma_{D0}$ is the uncertainty of the mean surface-weighted aerosol diameter ($\pm$ 0.5 % uncertainty), $mf$ is the mass fraction of 3-MGA in aqueous 3-MGA/AS droplet, $\sigma_{mf}$ is the uncertainty of mass fraction of solute ($\pm$ 5 % of $mf$ predicted by AIOMFAC-LLE for given RH), $\rho_0$ is the estimated aerosol density based on the volume additivity rule, $\sigma_{\rho_0}$ is the uncertainty of aerosol density determined using following equation.

$$\sigma_{\rho_0} = \rho_0{}^2 \frac{\rho*\rho_w + \rho_w*\rho_{AS} - 2*\rho*\rho_{AS}}{\rho*\rho_w*\rho_{AS}} * \sigma_{mf} \qquad (2)$$

where $\rho_w$ is the water density, $\rho$ is the density of 3-MGA, $\rho_{AS}$ is the density of AS.

**Reviewer's Comment #3**
In the discussion of diffusivity, the comparison is made for laboratory studies. How would this extrapolate to temperatures found in the atmosphere? Could we still anticipate that diffusion would not be limiting, especially given the lower OH radical concentrations?

**Authors' Response**
It acknowledges that when the ambient temperature decreases, the aerosol viscosity generally increases (everything else being equal). This would lead to a decrease in the diffusion rate of species from the bulk to the surface where oxidation preferentially takes place, and the overall rate of the oxidation will become more likely controlled by the diffusion. This is an expected temperature effect in the boundary layer (e.g. in the cold season or cold climates). However, in the context of vertical air motions (e.g. when air parcels rise adiabatically), we expect that a decrease in temperature will be accompanied by changes in RH; in the case of adiabatic ascent RH tends to increase. This in turn would potentially limit the increase in viscosity of hygroscopic aerosols or even lower it while RH remains high (Gervasi et al., 2020).

To investigate the effect of gas-phase OH radical concentrations, [OH], on the rate of oxidation, we could determine the characteristic time between two successive collision events between gas-phase OH radical and the aerosol surface, $\tau_{coll}$, as follow (Chim et al., 2018):

$$\tau_{coll} \cong \frac{4}{[OH] \, \overline{c_{OH}} \, A} \qquad (3)$$

where $\overline{c_{OH}}$ is the mean thermal velocity of gas-phase OH radicals, and $A$ is the surface area of the droplets. From Eqn. 3, $\tau_{coll}$ is larger at a lower gas-phase OH radical concentration. This would suggest that the species would have more time to diffuse to the aerosol surface for oxidation. Hence, as commented by the reviewer, the overall rate of the oxidation becomes less likely limited by the diffusion at lower gas-phase OH radical concentrations.

We have discussed the potential effect of temperatures and gas-phase OH radical concentrations on the heterogeneous reactivity in the revised manuscript.

Line 478, "It also notes that ambient gas-phase OH radical concentration is lower than that used in this study. This suggests that the species would have more time to diffuse to the aerosol surface for oxidation. Hence, the overall rate of the oxidation will be less likely limited by the diffusion at lower gas-phase OH radical concentrations in the atmosphere."

Conclusion, Line 579, "On the other hand, our results show that aqueous organic–inorganic droplets with more hydrophilic organic compounds (e.g. 3-MGA) may not necessarily experience diffusion limitation during heterogeneous OH oxidation, even when phase-separated. The overall heterogeneous reactivity is likely governed by the surface concentration of organic molecules at room temperature. It acknowledges that when the temperature decreases, the aerosol viscosity generally increases (everything else being equal). This would lead to a decrease in the diffusion rate of species from the bulk to the surface where oxidation preferentially takes place, and the overall rate of the oxidation will become more likely controlled by the diffusion. This is an expected temperature effect in the boundary layer (e.g. in the cold season or cold climates). However, in the context of vertical air motions (e.g. when air parcels rise adiabatically), a decrease in temperature will be accompanied by changes in RH; in the case of adiabatic ascent RH tends to increase. This in turn would potentially limit the increase in viscosity of hygroscopic aerosols or even lower it while RH remains high (Gervasi et al., 2020). Overall, this work further emphasizes that the effects of phase separation and potentially distinct aerosol morphologies add further complexity to the quantitative understanding of the heterogeneous reactivity of organic compounds in aqueous organic–inorganic droplets in the atmosphere, motivating further experimental and process modeling studies for a variety of aerosol systems."

**Reviewer's Comment #4**
The kinetics were tracked by looking at the loss of the parent signal, and the same products appear to be formed in the experiments. However, the intensities of these products have some apparent differences in Figure S4. Was there any correlation of product ion signals to the decay rate of the parent ion? Either in terms of the relative intensities between C6H9O5- or C6H7O5- or the total product ion signal?

**Authors' Response**
Thanks for the comment. We introduced a figure to show the change in the relative abundance of the two major functionalization products (alcohol product, $C_6H_9O_5^-$ and carbonyl product, $C_6H_7O_5^-$) as a function of OH exposure at different RH. As shown in Figure R1, the relative abundance of these two products increases with increasing OH exposure and does not significantly vary with the RH. To represent the correlation of major product ion signal to the decay of parent ion (i.e. 3-MGA), we calculated the change in relative abundance of the major product ion (i.e. $C_6H_9O_5^-$ or $C_6H_7O_5^-$) relative to that of parent ion (i.e. 3-MGA, $C_4H_9O_4^-$) at a given OH exposure as follow

$$Ratio = \frac{\Delta\,[product\ ion]}{\Delta\,[parent\ ion]} \qquad (4)$$

Table R1 shows that at the maximum OH exposure, the ratio for $C_6H_7O_5^-$ ranges from 0.23±0.06 to 0.33±0.11, while the ratio for $C_6H_9O_5^-$ ranges from 0.58±0.18 to 0.70±0.21 over the experimental RH. These results suggest that considering the uncertainties, the formation of the two major products does not strongly depend on the RH.

[Figure]

[Figure]

**Figure R1.** The relative abundance of carbonyl functionalization product ($C_6H_7O_5^-$, left panel) and alcohol functionalization product ($C_6H_9O_5^-$, right panel) under different relative humidities indicated by curve colour (legend) as a function of OH exposure.

**Table R1**. The change in relative abundance of the major product ion ($C_6H_9O_5^-$ and $C_6H_7O_5^-$) relative to that of parent ion at the maximum OH exposure at different RH

| RH (%) | 88 | 85 | 80 | 75 | 70 | 65 | 60 | 55 |
|---|---|---|---|---|---|---|---|---|
| Ratio for $C_6H_7O_5^-$ | 0.25±0.08 | 0.32±0.10 | 0.29±0.08 | 0.23±0.06 | 0.25±0.08 | 0.25±0.08 | 0.33±0.11 | 0.31±0.09 |
| Ratio for $C_6H_9O_5^-$ | 0.64±0.22 | 0.70±0.21 | 0.68±0.20 | 0.61±0.17 | 0.67±0.21 | 0.69±0.23 | 0.58±0.19 | 0.58±0.18 |

**Reviewer's Comment #5**
Figure 1 is not interpretable in black and white, I suggest a different color scheme, or more gradation.

**Authors' Response**
Thanks for the suggestion. We have revised the color scheme for Figure 1.

[Figure]

**Reviewer's Comment #6**
What are the error bars on Figure 5a (how are they estimated)? Are there error bars that can be applied to Figure 5b?

**Authors' Response**

In Figure 5a, the error bar for the x-axis represents the calculated error of OH exposure. The OH exposure, defined as the product of gas-phase OH radical concentration, [OH], and the particle residence time, t, was determined by measuring the decay of the hexane concentration (Smith et al., 2009):

$$OH\ exposure = -\frac{\ln([Hex]/[Hex]_0)}{k_{Hex}} = \int_0^t [OH]dt \qquad (5)$$

where [Hex] is the hexane concentration leaving the reactor after oxidation, [Hex]$_0$ is the initial hexane concentration before oxidation, and $k_{Hex}$ is the second-order rate constant of the gas-phase OH−hexane reaction). Based on Eqn.6 and the error propagation rule, the uncertainty for OH exposure, $\sigma_{OH\ exp}$, was derived from Eqn.7:

$$\sigma_{exp} = 0.005\ (OH\ exposure)\sqrt{\left(16 + \frac{2}{(OH\ exposure \times k_{Hex})^2}\right)} \qquad (6)$$

where 0.005 is the precision of the hexane concentration measurement (0.5 % of the reading). The error for the parent decay, $\sigma_{\frac{I}{I_0}}$, is determined from the following equation when the uncertainty of ion signal intensity was assigned to be 0.1 %:

$$\sigma_{\frac{I}{I_0}} = \frac{I}{I_0} \times 0.1 \times \sqrt{2} \qquad (7)$$

where $I$ is the signal intensity of 3-MGA at a given OH exposure and $I_0$ is the signal intensity before oxidation. In Figure 5b (right panel), the x-axis represents the relative humidity (RH), which has been measured by a calibrated RH and temperature sensor. The uncertainty of RH was estimated to be ± 1.5 % RH. The uncertainties of derived effective OH rate constants have been given in **Table 1**. The uncertainties have been included in Figure 5b (right panel) but they appear to be not obvious due to their smaller values. We have revised the Figure 5b (right panel) for better illustration.

[Figure]

References

Chim, M. M., Lim, C. Y., Kroll, J. H. and Chan, M. N.: Evolution in the reactivity of citric acid toward heterogeneous oxidation by gas-phase OH radicals, ACS Earth Space Chem., 2, 1323–1329, 2018.

Nah, T., Chan, M., Leone, S. R. and Wilson, K. R.: Real time in situ chemical characterization of submicrometer organic particles using direct analysis in Real Time-Mass Spectrometry, Anal. Chem., 85, 2087–2095, 2013.

Smith, J. D., Kroll, J. H., Cappa, C. D., Che, D. L., Liu, C. L., Ahmed, M., Leone, S. R., Worsnop, D. R. and Wilson, K. R.: The heterogeneous reaction of hydroxyl radicals with sub-micron squalane particles: a model system for understanding the oxidative aging of ambient aerosols, Atmos. Chem. Phys., 9, 3209–3222, 2009.